# Nutritional Implications of Bariatric Surgery on Pregnancy Management—A Narrative Review of the Literature

**DOI:** 10.3390/medicina59101864

**Published:** 2023-10-19

**Authors:** Anna Różańska-Walędziak, Maciej Walędziak, Anna Mierzejewska, Ewa Skopińska, Malwina Jędrysik, Beata Chełstowska

**Affiliations:** 1Department of Human Physiology and Pathophysiology, Faculty of Medicine, Collegium Medicum, Cardinal Stefan Wyszynski University in Warsaw, 01-938 Warsaw, Poland; aniaroza@tlen.pl (A.R.-W.); a.mierzejewska@uksw.edu.pl (A.M.); e.skopinska@uksw.edu.pl (E.S.); 2Department of General, Oncological, Metabolic and Thoracic Surgery, Military Institute of Medicine—National Research Institute, Szaserów 128 St., 04-141 Warsaw, Poland; 3Department of Biochemistry and Laboratory Diagnostics, Faculty of Medicine, Collegium Medicum, Cardinal Stefan Wyszynski University in Warsaw, 01-938 Warsaw, Poland; m.jedrysik@uksw.edu.pl (M.J.); b.chelstowska@uksw.edu.pl (B.C.)

**Keywords:** nutrition, pregnancy, bariatric surgery, micronutrient deficiencies, vitamin deficiencies, anemia

## Abstract

One in three women of reproductive age is obese. The mainstay treatment for obesity is bariatric surgery, and the following weight reduction results in a decrease in pregnancy adverse effects, including gestational diabetes mellitus, pregnancy-induced hypertension, and macrosomia. However, nutritional and vitamin deficiencies due to changes in the gastrointestinal tract after bariatric surgery are associated with an increase in the risk of fetal growth retardation and small for gestational-age neonates. The purpose of this review was to analyze the available recent literature on the subject of the management of pregnancy after bariatric surgery. We searched for available articles from 2007 to 2023 and chose articles of the greatest scientific and clinical value. Micronutrient, vitamin, and protein supplementation is recommended in the prenatal period and throughout the pregnancy. It is advised that pregnant women with a history of bariatric surgery should be provided with regular specialist dietary care. There is still a lack of recommendations about the optimum gestational weight gain after different types of bariatric surgery and for patients of different metabolic statuses. Women of reproductive age undergoing bariatric procedures should be provided with appropriate counseling about adequate contraception, the recommended time-to-conception interval, and the positive and negative influence of bariatric surgery on perinatal outcomes.

## 1. Introduction

Obesity is an important health problem worldwide, and the number of obese patients has doubled over the last two decades, being present in one in three women of childbearing age. Obesity is associated with numerous co-morbidities, including diabetes mellitus, hypertension, and obstructive sleep apnea, but also negatively influences fertility in both sexes and maternal and fetal outcomes. Obesity in pregnancy increases the risk of gestational diabetes mellitus (GDM), pregnancy-induced hypertension (PIH, medically indicated induction of labor, prolonged labor, vacuum delivery, planned cesarean sections, congenital anomalies, and LGA (large for gestational age) infants [1]. Obesity is not only a risk factor for hypertension in pregnant women but also preeclampsia and eclampsia. Bariatric surgery (BS) is the mainstay of the treatment of obesity, with the most effective long-term results in weight loss and reduction of obesity-related co-morbidities. BS should be complemented with lifestyle changes, including adequate diet and physical activity. As up to 80% of BSs are performed in women, half of them of childbearing age, it is important to follow the recommendations both during the pre-conception period and throughout the pregnancy [2,3]. There is no official consensus yet for the type of bariatric procedure indicated as the optimum one for women of reproductive age with respect to potential future pregnancy, though sleeve gastrectomy (SG) is associated with fewer nutritional deficiencies and surgical complications and is often a procedure of choice in women of childbearing age for bariatric surgeons [4].

Although the positive impact of BS on weight is undeniable, bariatric procedures influence dietary and nutritional intake [5]. BS leads to different macro- and micronutrient deficiencies, so adequate supplementation is of extreme importance, and patients are advised to take vitamin and mineral supplements for life [6]. Supplementation is of extreme importance, especially during pregnancy, when the maternal and fetal needs for vitamins and minerals are higher than outside pregnancy, and deficiencies can lead to more adverse effects, negatively influencing both maternal and fetal outcomes. Different types of bariatric procedures may have different effects on nutrient absorption and metabolism [7]. Roux-en-Y gastric bypass (RYGB) is more of a malabsorptive surgery (MS), and SG mostly represents restrictive surgery (RS); however, SG also has endocrine and metabolic effects [8]. Women of reproductive age with a history of BS should be monitored for nutritional status and potential deficiencies before conception, throughout pregnancy, and post-partum [9].

The purpose of our study was to present available up-to-date guidelines for the management of pregnancy after bariatric surgery, with special recognition of recommendations for nutrition and dietary supplementation.

## 2. Materials and Methods

The PubMed MEDLINE, Scopus, and Cochrane scientific databases were searched for original articles on the subject of management of pregnancy after bariatric surgery that had been published between 2007 and 2023. The research was performed by three independent researchers under the supervision of two senior researchers, and articles were retrieved using Google Scholar. The terms that were searched for were: “nutrition in pregnancy after bariatric surgery”, “supplementation in pregnancy after bariatric surgery”, “vitamin deficiencies in pregnancy after bariatric surgery”, “diabetes mellitus in pregnancy after bariatric surgery”, and “surgical complications in pregnancy after bariatric surgery”. Additional manual searches were conducted for the lists of references found according to the article’s novelty, quality, and clinical relevance. The review was divided into chapters, including the most common nutrient, micronutrient, and vitamin deficiencies, dietary supplementation, optimum time-to-conception after bariatric surgery, gestational weight gain, diagnosis of diabetes mellitus, pregnancy outcomes, and surgical complications.

## 3. Results and Discussion

### 3.1. Most Common Nutrient, Micronutrient, and Vitamin Deficiencies in Pregnancies after Bariatric Surgery

Coupaye et al. analyzed 123 s-trimester pregnancies and found disparities in the distribution of micronutrient and vitamin deficiencies after different types of BS. The caloric intake did not differ between SG and RYGB groups, but the protein intake was lower after SG. Women after RYGB were more likely to take multivitamins and other dietary supplements. Hemoglobin, calcium, and vitamin E levels were lower in the RYGB group, and there was a higher risk of vitamin B12 deficiency. Additionally, the RYGB group had higher levels of parathyroid hormone. There were no differences found between the groups in the serum albumin levels and urea excretion; the selenium and iron deficits were similar in both groups. Low protein and iron levels were correlated with impaired intrauterine fetal growth [10].

Rottenstreich et al. performed a systematic review of 27 articles that comprised 2056 pregnancies after BS. The reported deficiencies were common and included vitamins A, B1, B6, B12, C, D, and K, iron, calcium, selenium, and phosphorous. The adverse effects reported in the studies were anemia due to iron and B12 deficiency, urinary tract infections due to vitamin deficiencies, and night blindness due to vitamin deficiency. Iron, folate, and vitamins B1, B12, and D are found both after restrictive and malabsorptive procedures, while vitamins A, K, and E, calcium, zinc, and copper deficiencies are found mostly after malabsorptive procedures. The prevalence of anemia in pregnant post-bariatric patients differed between the studies, from 17% up to 77% of patients. A total of 10% to 16% of patients had to be treated with intravenous iron supplementation throughout pregnancy, and 3% to 17% required blood transfusion. A higher prevalence of anemia was correlated with a longer time-to-conception (TTC) interval. Vitamin A deficiency was found in up to 90% of post-bariatric pregnant patients, and night blindness was found in 57% up to 87% of patients [2].

Anemia in pregnant patients after BS mostly results from vitamin B12, iron, or folate deficiency, but unexplained anemia that is not treated by classic supplementation should be investigated, as it can be a symptom of copper or zinc deficiency [6]. As zinc and copper share the same absorption pathway, a zinc-induced copper deficiency can occur. The supplementation of zinc and copper should maintain the optimum ratio of 8–15 mg of zinc per 1 mg of copper [6].

Micronutrients and vitamin deficiencies after BS do not diminish the importance of breastfeeding. Women after BS should receive appropriate supplementation and should be encouraged to breastfeed [11]. Mineral and vitamin serum levels should be regularly monitored, and deficiencies should be adequately corrected. The available research shows that there is no evidence of worsened quality of milk or long-term adverse effects in children of mothers after BS. Additionally, breastfeeding can reduce the risk of obesity in children of mothers after BS [12]. However, BS was not found to have had a positive influence on the risk of offspring obesity. Gothelf et al. analyzed the health of the offspring of 1086 women who had children both before and after BS, with a follow-up of 18 years, and found there were no differences in the risk of obesity or prevalence of co-morbidities [13].

### 3.2. Dietary Supplementation in Pregnancies after Bariatric Surgery

Multivitamin and mineral supplementation prior to conception and throughout pregnancy is recommended by all international guidelines considering pregnancy after BS [14]. The multivitamin supplement should contain at least the following amounts of micronutrients and vitamins: folic acid—0.4–1 mg, iron—45–60 mg (18 mg after adjustable gastric band (AGB)), thiamine—12 mg, vitamin E—15 mg, copper—2 mg, zinc—15 mg, selenium—50 μg, and beta-carotene (vitamin A)—5000 IU [15].

Folic acid supplementation after BS is still being discussed. For pregnant women who remain obese after BS, the recommended daily dose of folic acid before conception and in the first trimester is 4–5 mg. However, it is not recommended for the whole population after BS because folic acid deficiency is less common than vitamin B12, especially in patients after MS, and over-supplementation of folic acid can mask the symptoms of vitamin B12 deficiency despite augmenting neurological defects. Most studies do not show a higher prevalence of folate deficiency in pregnant patients after BS than in the general pregnant population [2]. Mead et al. showed in their study of 113 pregnant women who had had three types of BS that folate serum levels were increased in all patients apart from two who admitted no compliance to recommended supplementation. High folate levels in patients probably result from the received supplementation [16]. Some studies indicate that continuous, adequate supplementation of standard multivitamin supplements of multivitamins specially developed for post-bariatric patients before and throughout pregnancy reduces the rate of observed deficiencies. In 197 pregnancy cases after RYGB, observed at a bariatric expertise center in the Netherlands, iron serum levels remained stable, folate, vitamin D, and vitamin B12 levels increased, and only hemoglobin and calcium levels were reported to have been decreased [17]. Eissa et al. analyzed a group of 245 pregnant women after SG and found an incidence of anemia of 78.8% [18]. Vitamin B12 should be supplemented in all patients after BS, both during the pre-conception period and throughout pregnancy, either as an intramuscular depot injection of 1 mg every 3 months or orally (1 mg daily). The way of supplementation can be chosen according to individual preference and effectiveness [15].

Iron deficiency anemia is defined by hemoglobin levels below 105 g/L and ferritin levels below 30 μg/L. The minimum recommended daily dosage of iron is 45 mg of elemental iron (18 mg after AGB); however, it has to be adjusted according to the patient’s ferritin and hemoglobin levels [10]. If oral supplementation does not result in elevation of ferritin and hemoglobin levels or anemia is refractory, intravenous infusion should be administered. The dose is calculated based on body weight and hemoglobin level; in most cases, 1000 mg is sufficient to treat anemia, and it takes 6 to 8 weeks for the hemoglobin serum levels to normalize [19]. Intravenous iron infusions are not recommended in the first trimester due to a lack of data [20].

The retinol form of vitamin A is teratogenic and, therefore, should be avoided during pregnancy. Vitamin D should be maintained at a level of at least 50 nmol/L (or 75 nmol/L depending on different recommendations), and the serum parathyroid hormone should be within normal limits, with a possible additional supplementation of calcium. Other studies recommend supplementation of calcium in all pregnant patients after BS [14]. There is a basic risk of thiamin deficiency after BS, which can be augmented in patients who suffer from prolonged vomiting after BS, especially if it is aggravated in pregnancy. In case of prolonged vomiting, each patient should be treated for potential thiamine deficiency to prevent irreparable brain damage [6]. Thiamin is involved in the synthesis of myelin, the formation of mitochondrial and synaptic membranes, and the process of fetal neural and brain development [8].

The serum levels of the following parameters should be checked at least once a trimester: full blood count, including hemoglobin, ferritin, transferrin, folate, vitamin B12, vitamin D, calcium, magnesium, phosphate, and parathyroid hormone, followed by vitamin A, prothrombin time, INR, vitamin K1, protein, albumin, and renal and liver function tests. If any deficiencies are found, the intensity of exams should be adjusted for the optimum monitoring [15].

The diet of pregnant women should be planned with a bariatric dietitian and specialistic dietary care should be maintained throughout the pregnancy. The diet should be prepared individually based on the type of BS, TTC, pre-pregnancy body mass index (BMI), gestational weight gain (GWG), physical activity, and individual food preference and tolerance. General rules include small but frequent meals, long chewing, drinking between (not during) meals, and avoiding simple carbohydrates. According to some studies, pregnant patients who have had BS may require an additional 200 kcal per day [21]. GWG is the preferred indicator for the adequacy of calorie intake [19]. Women with a history of BS are often afraid of weight regain during pregnancy, and their daily caloric intake is insufficient; therefore, it is crucial to provide them with specialist dietary and psychological care. The importance of psychological assistance is also important because depressive symptoms in pregnant women after BS affect approximately one-third of patients [22]. Hedderson et al. presented a telephonic nutritional management program for pregnant women after BS. In a group of 1142 participants, there was a lower risk of preterm birth (aRR 0.48, 95% CI 0.35–0.67), preeclampsia or gestational hypertension (aRR 0.43, 95% CI 0.27–0.69 and RR 0.62, 95% CI 0.41–0.93, respectively), and level 2 or 3 neonate NICU admission (aRR 0.61, 95% CI 0.39–0.94 and aRR 0.66, 95% CI 0.45–0.97, respectively [23]. Araki et al. analyzed the effects of personalized nutrition counseling on pregnancy outcomes and found that regular professional dietary appointments were associated with improved nutrient intake and better food quality habits [24]. The recommended protein intake is at least 60 g daily, while the optimum is 1.5 g per kg per day, and in some cases, it needs to be even higher. If the necessary protein intake is not fulfilled by the diet, it has to be supplemented [15]. Protein malnutrition is defined by serum albumin <25 g/L and is more common after MS, although there is a risk of protein malnutrition after both RS and MS [21].

### 3.3. Optimum Time-to-Conception Interval after Bariatric Surgery

International guidelines clearly state that it is not recommended to begin pregnancy in a period of rapid weight loss after BS. The length of the catabolism period differs between the patients, depending mostly on the pre-surgery BMI and the type of surgery. In most patients, weight stabilization is achieved a year after surgery. Conception is recommended after at least 2 months of plateau of body weight. The recommended TTC differs between guidelines from 1 to 2 years after BS [2]. The American Society for Metabolic and Bariatric Surgery, the Obesity Society, and the American Association of Clinical Endocrinology suggest a TTC interval of 12 to 18 months [25]

During the rapid weight loss period, there is a high risk of nutritional deficiencies and, subsequently, the impairment of fetal growth or spontaneous abortion. Different studies confirm that pregnancies started under 12 months after BS are associated with a higher risk of inadequate gestational weight gain (GWG), which, in consequence, can negatively influence fetal growth [26,27,28]. A study by Solaiman et al. in a group of 158 pregnant patients after BS indicated that there was a negative relationship between shorter TTC interval and GWG (*p* = 0.002) and lower birth weight and inadequate GWG (*p* = 0.03); however, the study showed no significant correlation between TTC and PIH, and GDM and neonatal outcomes [26]. In a recent Chinese study, Wang et al. found that the spontaneous abortion rate was higher when TTC was lower than 2 years, compared to a TTC interval of 2 years and more (*p* = 0.04) [29].

However, there are studies contraindicating the necessity of maintaining a TTC interval after BS. Froylich et al. analyzed 95 pregnancies after laparoscopic sleeve gastrectomy (LSG), comparing three groups with TTC intervals of < 12 months, 12–24 months, and >24 months. They found no significant differences in pregnancy course or complications and the neonates’ weight. The possible bias of the study was the size of the groups and the history of only one type of BS, but the results are interesting and to be further studied [30]. The study group by White et al. comprised 135 women after SG or RYGB. The researchers did not find any statistically significant differences in the prevalence of perinatal outcomes, including small for gestational age neonates (SGA) or neonatal intensive care unit (NICU) admissions [31]. TTC can adversely affect the long-term effects of weight loss after BS. Nijland et al. compared a group of women who had become pregnant within a TTC interval of less than 12 months, 12 to 24 months, and more than 24 months after BS. Weight outcomes were measured until 5 years after BS. The study indicated that women who had become pregnant within 12 months after BS achieved lower estimated weight loss % (EWL%) than those who had become pregnant after 12 months and more after BS (*p* = 0.02), with no differences between groups with a TTC interval of 12 and 24 months and >24 months [32].

A study in a cohort of 1464 patients after BS who gave birth in the United States between 2011 and 2017 found that more than 85% of births occurred within 21 months after BS. Chao et al. found that delivery during the 2 years after BS was not associated with an increased risk of surgical reinterventions [33]. Additionally, the surgery-to-delivery interval was found to have had no significant impact on the health outcome of children in the first year of life [34]. An important issue to be remembered is that women will become pregnant with a shorter TTC interval. It is important to provide them with non-judgmental and supportive care to avoid further consequences of lack of compliance and minimize the risk of poor perinatal outcomes [35].

### 3.4. Gestational Weight Gain and Bariatric Surgery

Gestational weight gain is positively associated with the fetal growth [10,36]. There are no specific recommendations for the optimum GWG in patients after BS, and in most studies, the norms used are those for the general pregnant population according to pre-pregnancy BMI [37,38]. Excessive weight gain in pregnancy after BS can lead to post-partum weight retention; insufficient GWG can result in fetal growth retardation [39,40].

In a study by Heusschen et al., GWG was analyzed in correlation with time to conception after BS. Patients were divided into three groups: early (less than 12 months after BS), middle (12 to 24 months after BS), and late (more than 24 months after BS). Pregnancy in the early group was associated with lower gestational weight gain and lower neonatal birth weight. Inadequate gestational weight gain was associated with lower gestational age at delivery (266.5 ± 20.2 days vs. 273.8 ± 8.4 days, *p* = 0.002) [28]. These results are confirmed in a study by Grandfils et al., who analyzed 337 pregnancies after BS and found that insufficient GWG was a risk factor for small for gestational age (SGA) neonates when compared to excessive GWG (OR, 1.96, 95% CI 1.04–3.68), preterm labor (OR, 4.13, 95% CI 1.84–9.24), and preterm delivery (OR, 6.40, 95% CI 2.41–17.0) [27]. Contradictory, Sancak et al. analyzed 119 pregnancies after LSG. The difference between Sancak’s study and the others previously mentioned was the type of surgery performed, with only LSG used in Sansack’s study and different types of restrictive and malabsorptive surgery used in Heusschen’s and Grandfils’ studies, which might have affected the results.

### 3.5. Diagnosis of Gestational Diabetes Mellitus in Pregnancy after Bariatric Surgery

The incidence of GDM in pregnancies following BS has been shown to be lower, with one reason being the improvement in insulin resistance. The risk of developing GDM remains higher in women who remain obese after BS. An oral glucose tolerance test (OGTT) is not recommended in pregnancies after BS because there is concern about its tolerability, safety, and accuracy [40]. Fast intake of simple carbohydrates in patients after BS, especially MS, leads to hyperinsulinemia and the following hypoglycemia, called postprandial hyperinsulinemic hypoglycemia (PHH), which clinically shows as late dumping syndrome and can be harmful both for the mother and the fetus. However, PHH of even 40 mg/dL can remain asymptomatic and, therefore, dangerous for both the mother and the child [41]. There are limited possibilities of treatment for PHH, and the mainstay of treatment is dietary restrictions. Simple carbohydrates should be excluded, fluid intake separated from meals, and the content of protein and fat should be increased. Some additives can be used to slow gastric emptying, including starch, pectin, guar gum, and maltodextrin. In the case of a PHH acute event, there are case reports of using acarbose, but no studies have been conducted in larger groups [41]. Among the other measures used in treating hypoglycemia in pregnancies after BS, there is also the possibility of continuous overnight intravenous dextrose infusion [42].

According to recent studies, there is a correlation between hypoglycemic events in pregnancies after BS and decreased birth weight of neonates. Women who had hypoglycemia during OGTT had a higher risk of repetitive hypoglycemic events in pregnancy and, therefore, intrauterine growth retardation (IUGR), and there was an association found between the maternal glucose nadir during OGTT and fetal growth restriction [43]. The alternative forms of screening for GDM in patients after BS are seven-point capillary blood glucose monitoring or continuous glucose monitoring (GCM) for one week between the 24th and 28th weeks of pregnancy. As there are no specific norms for glucose levels in pregnancies after BS, it is recommended to use the general international norms for pregnancy—95 mg fasting glucose and 140 mg glucose level one hour after a meal [40]. Studies analyzing CGM results in pregnancies after BS show that an early glucose peak and rapid decrease in postprandial glucose levels, as well as lower fasting glucose levels than in the general pregnant population, are characteristic of pregnant patients after BS [44].

### 3.6. Pregnancy Outcomes after Bariatric Surgery

According to most studies, patients after BS have a reduced risk of fetal macrosomia, PIH, preeclampsia, and GDM, whereas the risk of IUGR and SGA is increased [45,46,47]. In a recent study by Boller et al. in a cohort of 1591 pregnancies after BS, pregnancy after BS was associated with a decreased risk of GDM or impaired fasting glucose (23.5% vs. 35.0%, aRR 0.73, 95% CI 0.66–0.80), preeclampsia (7.5% vs. 10.2%, aRR0.72, 95% CI 0.60–0.86), and LGA (10.6% vs. 19.9%, aRR 0.56, 95% CI 0.48–0.65) and an increased risk of SGA (10.9% vs. 6.6%, aRR 1.51, 95% CI 1.28–1.78) [48].

A systematic review and meta-analysis by Zakhter et al. analyzed adverse perinatal outcomes of 14,880 pregnancies after BS. A history of BS (all types combined) increased the risk of perinatal mortality (OR 1.38, 95% CI 1.03–1.85, *p* = 0.031), congenital anomalies (OR 1.29, 95% CI 1.04–1.59, *p* = 0.019), preterm birth (OR 1.57, 95% CI 1.38–1.79, *p* < 0.001), and NICU admission (OR 1.41, 95% CI 1.25–1.59, *p* < 0.001). After RYGB, but not AGB, odd ratios for SGA were higher (OR 2.72, 95% CI 2.32–3.20, *p* < 0.001), and for LGA, ratios were lower (OR 0.24, 95% CI 0.14–0.41, *p* < 0.001). The incidence of post-term birth was decreased after BS (OR 0.46, 95% CI 0.35–0.60, *p* < 0.001). The birth weight of neonates born to mothers after BS (all types combined) was more than 200 g lower than that of those born to mothers without prior BS (weight mean difference −242.42 g, 95% CI −307.43 to −177.40 g, *p* < 0.01) [49].

BS is known to be associated with a higher risk of IUGR. There is a correlation between maternal protein deficiency, more common after BS than in the general population, and the rate of incidence of IUGR and small for gestational age (SGA) neonates. The risk of IUGR in most studies is higher in patients after MS [46]. There is conflicting data about the type of BS with a higher prevalence of protein deficiency; however, one of the main reasons for the protein deficiency is a lack of patients’ compliance and daily intake of proteins lower than the recommended 60 g [10]. Additionally, improved insulin resistance levels after BS can lead to an increased risk of SGA, as insulin resistance might be insufficient in the third trimester to provide enough glucose for the fetus [15].

Most studies indicate a 2–3-fold decrease in the incidence of GDM after BS, with reduced insulin resistance as an explanation [50]. There are some studies indicating that the risk of GDM after BS might be increased, not decreased. Sesilia et al. analyzed a group of 314 women after BS, in whom they found an increased risk of GDM when compared to the non-bariatric group with adjusted BMI (*p* = 0.018) and SGA infants (*p* < 0.001) [51].

There are conflicting findings about the rate of CS, though most studies indicate an increase in the rate of CS after BS. In a cohort of 53,950 pregnant patients after BS, Youssefzadeh et al. found that the rate of CS was higher in women with a history of CS than in women without prior BS (47.0% vs. 32.2%; *p* <0.001) [52]. However, some authors state that the rate of CS decreases after BS when compared to the obese population. Savastano et al. found a significant reduction in the CS rate after BS (20% vs. 68.3%; *p* = 0.006) [45].

### 3.7. Surgical Complications in Pregnancy after Bariatric Surgery

There are only case series and small cohort studies available on the subject of surgical complications after BS. The risk of serious complications that require surgical intervention during pregnancy is incomparably higher after RYGB than after SG. Pregnancy can increase the risk of internal herniation due to an increase in intra-abdominal pressure and displacement of the small intestine as the pregnant uterus grows. Internal herniation can lead to obstruction of the small intestine and the following necrosis that can be avoided only with early surgical intervention. The rate of incidence of internal herniation after RYGB is reported as 2–12% [53].

Clinical presentation of internal herniation can be mistaken for typical symptoms in pregnancy: abdominal pain, in most cases, postprandial, nausea, and vomiting. It is of extreme importance to increase patients’ awareness of the risk of this complication. A patient with the symptoms mentioned above that appear in pregnancy after BS should report herself immediately to an emergency ward, as there is a risk of maternal and fetal death, especially more than 48 h after the beginning of symptoms. The patient should report to a center with both bariatric care and maternal and neonatal care, as it can reduce the rate of both maternal and neonatal mortality to 0% [54]. In any case of a pregnant patient after RYGB presenting with abdominal pain, nausea, and vomiting, she should be examined by a bariatric surgeon. The optimum method of confirming the diagnosis is magnetic resonance imaging (MRI); computed tomography cannot exclude internal herniation and is disadvised in pregnancy due to ionizing radiation. If MRI is not available, a diagnostic laparoscopy should be the method of choice [54].

Another type of surgical complication in pregnancy after BS is gastric band slippage, which can take place in up to 12% of pregnancies after gastric banding. The intra-abdominal pressure is increased by the pregnancy itself and can be aggravated by vomiting, the result of which can be gastric band slippage.

### 3.8. Women’s Awareness about Gynecological and Obstetric Care after Bariatric Surgery

Rapid and sustained weight loss after BS can lead to a sudden increase in fertility. A pregnancy is unadvised during the period of weight loss after BS for at least a year. Therefore, a female patient of reproductive age should receive contraceptive advice before and after BS. There is a low level of contraception counseling. A recent survey by Graham et al. asked 382 bariatric surgeons if they provided adequate information about contraception to reproductive-aged bariatric surgical patients. The response rate was low, at 17%; only 7% of bariatric surgeons reported having provided information about contraception to their patients [55]. Mengesha et al. have observed that even one perioperative contraception counseling has a positive effect on adequate contraception use. The study was conducted in a group of 363 women from the United States who had undergone bariatric surgery within 24 months prior to the interview. Three-quarters of women had preoperative counseling about contraception and pregnancy with their bariatric providers, though 41% of them would have preferred more counseling. Perioperative contraceptive or pregnancy discussions were independently associated with a higher rate of use of contraception after the surgery (OR 2.5; 95% CI, 1.5–4.3, *p* < 0.001) [56].

There is also a need to inform women about the positive and negative influence of BS on perinatal outcomes. Patients should be informed about the possible nutritional deficiencies, the importance of adequate diet and supplementation, and their influence on fetal growth, as patient compliance is crucial to optimum pregnancy care. In a study by Assaf et al., conducted in a group of 183 women after BS, aged 15–49, more than half of the participants were not aware of the influence of BS on perinatal outcomes [57].

## 4. Limitations of the Study

One of the main limitations of this study was that there were different lifestyle factors of the participants associated with their nationality, culture, religion, sociodemographic conditions, and their influence on patients’ nutrition. Additionally, there was no data considering the diet of the participants, and only a few studies presented information about the characteristics of the supplementation received by the participants. It is also impossible to determine the patient’s compliance, both according to diet and supplementation. Apart from the information about the possible masking of vitamin B12 anemia symptoms by overdosage of folic acid, we have not found any information about the risks and adverse effects of supplementation.

## 5. Conclusions

Some questions remain unanswered. It is still to be established whether the minimum TTC interval of 12 months after BS to avoid adverse perinatal outcomes is necessary in all patients and after all types of surgeries, as the results differ between the studies. Most include more patients after RYGB, whereas SG is the most common bariatric procedure in Europe and leads to fewer micronutrients and vitamin deficiencies. As sleeve gastrectomy is associated with a lower risk of nutritional deficiencies and its consequences, including different types of deficiency, as well as surgical complications, we suggest it should be considered as the procedure of choice for women of childbearing age.

There is still a lack of recommendations about the optimum gestational weight gain after different types of BS and within different TTC intervals and BMI. More prospective studies in large groups are necessary to establish the prevalence of various micronutrient deficiencies after different types of surgeries and in different populations. There is a lack of adequate pre-surgery counseling about optimum contraception and the positive and negative influence of BS on perinatal outcomes for reproductive-age women considering both surgery and procreation.

## Data Availability

Not applicable.

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
