# Peer review of "Nutritional Implications of Bariatric Surgery on Pregnancy Management—A Narrative Review of the Literature"

_medicina, 2023, doi:10.3390/medicina59101864_

Round 1

Reviewer 1 Report

The authors have analyzed the available recent literature on management of pregnancy after bariatric surgery. The paper is written in poorly manner and cannot be accepted for publication. Following are major flaws:

·         The title of the paper does not match the text. The paper does not provide any guidelines. Infact they have focused on nutrition and bariatric surgery and its implications in pregnancy.

·         The result section is just compilation of various studies in different areas. There are no inferences on the reported studies.

·         Authors fail to identify the gaps between bariatric surgery and nutritional status and pregnancy.

·         Paper is written without any figures. It is too boring to read such papers and few figures should be added to increase the readability. Similarly, no tables are included.

    Language needs major corrections.

Author Response

Dear Reviewer 1,

Thank you for your review of our manuscript.

According to your indication, we corrected the title of our manuscript and removed the word ‘guidelines’, changing it into:  “Implications of nutrition after bariatric surgery on management of pregnancy - a review of literature”.

We tried to present information considering nutrition in pregnancy after bariatric surgery found in literature and our conclusions.

Reviewer 2 Report

The work makes a good impression and is written at a modern level. There are the following suggestions and recommendations for this work:

1. in the introduction, indicate that obesity is not only a risk factor for hypertension in pregnant women, but also a risk factor for preeclampsia and eclampsia

2.Edit the text and exclude paragraphs containing one small sentence (for example, "The retinol form of vitamin A is teratogenic and therefore should be avoided during pregnancy. ")

3. at the end of the article, provide information about the limitations of this study

Author Response

Dear Reviewer 2,

Thank you for your positive review of our manuscript.

Following your suggestion, we added that obesity is not only a risk factor for hypertension in pregnant women, but also a risk factor for preeclampsia and eclampsia.

We edited the text and excluded paragraphs containing short sentences, combining them into longer paragraphs.

According to your suggestion, we added paragraph about the limitations of the study.

Reviewer 3 Report

Revision and comments

1.       Line 17 to 20, this affirmation is incorrect, “The mainstay treatment for obesity is bariatric surgery and the following weight reduction results in decrease of pregnancy adverse effects due to obesity, including gestational diabetes mellitus, pregnancy-induced hypertension and macrosomia”.  There is sufficient evidence that the modification of healthy lifestyles and food education are the mainstay, since any treatment for obesity should not be based only on weight, but on its distribution and body fat so that any treatment is sustainable.

2.       Variables such as the amount of muscle mass, physical activity, anemia or history, number of pregnancies, extreme changes in weight and other risk factors that are specific to women must be considered.

3.       In lines 50 to 55, the authors mentioned that there is no official consensus yet for the type of bariatric procedure indicated as the optimal one for women of reproductive age in respect of potential future pregnancy; Perhaps the reflection should be whether the benefits of opting for bariatric surgery outweigh the risks and these be contrasted with the evidence available with nutritional and educational intervention, since the approach presented to BS is as an alternative and not as a complement to treatment. Therefore, it is suggested to expand the argumentation and justification.

4.       As an orthographic observation, place the “period” at the end of the parenthesis throughout the manuscript.

5.       In Lines 56 to 68, a discrepancy is observed since undeniable benefits are mentioned only on weight and with adverse effects for life such as the suggestion of taking vitamin D for life, leaving aside the accumulated evidence on the benefits of A nutritional intervention, on the other hand, does not specify the type of patients for whom the surgery is intended, that is, level of education, socioeconomic level, situation of vulnerability, type of health service (public or private).

6.       Although the authors point out that there is no official consensus yet for the type of bariatric procedure indicated as the optimal one for women of reproductive age in respect of potential future pregnancy, why not consider the evidence that exists for non-pregnant women to start from the bases and establish the risks or benefits.

7.       Regarding the selection of studies, why if evidence of the management of SB in pregnancy is sought, why were clinical trials or designs with greater power or level of evidence not selected.

8. Results are presented, but there is no discussion about it. For example, why address only a few articles on “SURGICAL COMPLICATIONS IN PREGNANCY AFTER BARIATRIC SURGERY”, when you can discuss and mention all the adverse events that were presented or reported during the entire review.

9. The authors give more value to weight and not to its composition, nor do they address changes in the lifestyle or type of diet that the patients followed.

10. The conclusion is limited in accordance with what is established in the objectives of the study and its justification. The discussion and conclusion need to be expanded further.

11. It is recommended to complement the review with a table with a list of studies.

12. The study is of interest but the probable biases or confounding variables are not addressed, such as diet, lifestyle, sociodemographic conditions of the participants, the countries from which the main studies were selected and finally, the study does not considers or discusses the study design, risks or adverse events found.

Author Response

Dear Reviewer 3,

Thank you for the review of our manuscript.

We underlined that bariatric surgery should supported by lifestyle changes, including adequate diet and physical activity.

We indicated the importance of variables such as the amount of muscle mass, physical activity, anemia or history, number of pregnancies, extreme changes in weight and other risk factors that are specific to women.

In the conclusions we added the suggestion of considering sleeve gastrectomy as a method of choice for women of reproductive age, as it as associated with a lower risk of nutritional deficiencies, including different types of anemia and lower risk of surgical complications. The available guidelines on the management of pregnancy after bariatric surgery were presented in the manuscript, however data is limited and differs between the studies. That is why we presented the issues to be resolved in the conclusions.

As it was suggested by the editor of the special edition of the journal, the manuscript was not supposed to be a systematic review, therefore we did not include the table with the list of studies.

Following your remark, we added a paragraph about the limitations of our study, including the probable biases like  as diet, lifestyle, sociodemographic conditions of the participants, the countries from which the main studies were selected and the risks or adverse events found.

Round 2

Reviewer 1 Report

Authors have not yet addressed the issues. No figures/tables have been added. No inferences have been drawn from the literature review.

Poor